# The Relationship between Soil Moisture and Soil Water Repellency Persistence in Hydrophobic Soils

**Mohamed Bayad** [1,2], **Henry Wai Chau** [1,*], **Stephen Trolove** [3], **Jim Moir** [1], **Leo Condron** [1] **and Moussa Bouray** [1,2]

1   Department of Soil and Physical Sciences, Lincoln University, Christchurch 7647, New Zealand; mohamed.bayad@lincolnuni.ac.nz (M.B.); Jim.Moir@lincoln.ac.nz (J.M.); Leo.Condron@lincoln.ac.nz (L.C.); moussa.bouray@lincolnuni.ac.nz (M.B.)
2   AgroBioSciences Program, Mohammed VI Polytechnic University, Benguerir 43150, Morocco
3   The New Zealand Institute for Plant & Food Research Limited, Private Bag 140, Havelock North 4157, New Zealand; stephen.trolove@plantandfood.co.nz
*   Correspondence: henry.chau@lincoln.ac.nz; Tel.: +64-033253607

**Abstract:** In this work, we modelled the response of soil water repellency (SWR) persistence to the decrease in moisture in drying soils, and we explored the implication of soil particle size distribution and specific surface area on the SWR severity and persistence. A new equation for the relationship between SWR persistence and soil moisture ($\theta$) is described in this paper. The persistence of SWR was measured on ten different hydrophobic soils using water drop penetration time (WDPT) at decreasing levels of gravimetric water content. The actual repellency persistence showed a sigmoidal response to soil moisture decrease, where $R_a(\theta) = R_p/1 + e^{\delta(\theta-\theta_c)}$. The suggested equation enables one to model the actual SWR persistence ($R_a$) using $\theta$, the potential repellency ($R_p$) and two characteristic parameters related to the shape of the response curve. The two parameters are the critical soil moisture $\theta_c$, where the $R_a$ increase rate reaches its maximum, and the parameter $\delta$ affecting the steepness of the curve at the inflexion point of the sigmoidal curve. Data shows that both soil carbon and texture are controlling the potential SWR in New Zealand pastures.

**Keywords:** soil water repellency; soil moisture; soil carbon; soil texture

## 1. Introduction

Soil water repellency refers to the inability of soils to absorb water. This phenomenon has been identified in different soils and climate combinations [1]. Theoretically, the origin of water repellency is due to hydrophobic materials coating the soil particles [2]. Doerr and Thomas [3] theorized that after the amphiphilic molecules have been separated from the mineral particles during the wetting of the soil (making the soil particles wettable), these molecules remain intact. When the soil moisture becomes low enough during the drying process, their polar groups re-associate and interact through the hydrogen bonds, forcing the molecules back into position with the polar heads attached to the mineral surface and the non-polar tails orientated outwards resulting in the reestablishment of hydrophobicity [3–8]. Therefore, soils express water repellency when moisture drops below a critical water content. This issue has serious implications in decreasing water infiltration [9], inducing surface runoff, nutrient losses through runoff [10] and causing preferential flow [11].

Soil water repellency can be characterized by two different criteria. Its severity or degree, which is an estimate of the soil surface tension, indicates the initial strength of the repellency between water and soil surface. The persistence is a measure of how long it takes to break down the repellent property after prolonged contact with water to make the soil wettable again. The severity of SWR can be estimated

using the molarity of ethanol drop (MED) [12], contact angle (CA) [13] or, sessile drop methods [14,15]. The persistence of soil water repellency (SWR) can be estimated using water drop penetration time (WDPT) [16]. The water drop penetration time approach involves placing a water drop on the soil surface and recording the time it takes to penetrate [16]. As water infiltrates the soil surface when the contact angle is less than 90°, WDPT measures the time taken by the contact angle to change from values >90° to 90° [16] or 0° if a complete drop penetration is considered [12].

The actual SWR level is tightly related to soil moisture. Soil water repellency as a function of θ, which is usually referred to as the SWR characteristic curve and has been investigated in different studies [17–19]. However, most of the previous work has focused on SWR severity and moisture relationship [17,18,20], and less effort has been made on SWR persistence dynamics. The characteristic curves of SWR (MED or CA) as a function of soil moisture (θ) have been studied in reclaimed and agricultural soil [17], pastures [18] and, forest ecosystems [20]. In a study of the relationship between the SWR severity and water content in natural, reclaimed and agricultural soils, Chau et al. [17] revealed that SWR severity showed different patterns in drying soils. Some soils showed a rapid decrease in the contact angle when θ increased; this indicated less severity, regardless of the initial repellency. Other soils showed a slow decrease in contact angle at higher water contents, which indicated high severity and the need for more water or surfactants for remediation [17]. The SWR severity characteristic curves are typically a unimodal curve where the contact angle increases with decreasing θ, reaches a peak and then decreases again [8]. Other studies showed bimodal response curves, sigmoidal or irregular patterns [17,18].

The SWR severity is a measure of the initial strength of the soil hydrophobicity and does not describe how soil behaves in prolonged contact with water. Thus, SWR measured with MED or CA, for example, does not permit the perception of how SWR influences natural hydrological processes. The parameters controlling the SWR persistence response to drying are to date, still poorly understood. The only study implicating modelling the persistence of SWR in drying soils was conducted on artificially hydrophobized soils [21]. Data from this study showed a typical sigmoidal curve of WDPT (θ) for sand and unimodal curves for finer textures. Artificially induced hydrophobicity has been used in many studies to understand the impact of water repellency on hydrological processes at the theoretical level. However, the stable characteristics of the artificial hydrophobic chemicals used for this purpose make it difficult to compare results with naturally hydrophobic soils. Natural SWR represents unique features because it results from a combination of natural hydrophobic materials of different sources [22]. Modelling the SWR persistence characteristic curves is urgently needed for naturally hydrophobic soils. This will help to understand and quantify the impact of SWR on runoff and nutrients losses to waterways in agroecosystems.

Characterization of actual SWR and water content relationship was assessed in field conditions [23,24], by adding water or drying soils in laboratory conditions [8,18,25]. Using air or oven-dry (65 °C) soils gives an estimate of the potential SWR, which is the highest level that SWR can reach when soil dries out completely [23,26]. Estimation of the potential SWR provides insight into the potential consequences of soil hydrophobicity in an eventual drought situation. However, only in-situ measurement at field moist conditions gives the actual SWR level [24]. Adding water would simulate the soil wetting phase but does not give information about SWR dynamics during the drying phase [21]. On a drying soil surface, the hydraulic potential is constantly changing, and hydrophobic compounds would not have the same behaviour toward soil minerals, as if they remained at constant water content for 48 h. Thus, understanding the SWR–moisture natural dynamics needs the closest possible scenario to field dry conditions. Moreover, the drying temperature has a significant effect on the reestablishment of SWR in sands [27]. The micromorphological investigation by Dekker et al. [27] showed that high drying temperatures caused an increase in the formation of organic materials coatings responsible for SWR. Hence, soil drying at 105 °C used in some studies can give an incorrect estimate of SWR. Air drying was suggested as a laboratory approach to study the SWR–moisture relationship in different studies [3].

Still, there was no systematic work on the effect of drying temperature on SWR reestablishment for different soil textures.

Throughout the literature, there have been substantial advances in understanding the soil properties affecting the potential SWR. There is a strong indication of organic compounds' implication on controlling the potential SWR [19,28]. However, a large body of research shows the implication of soil texture as another important factor controlling SWR levels. Although SWR can occur in a wide range of soil textures [26,29,30], sandy soils are more susceptible to coating by hydrophobic materials because of their low surface area (SA) [31]. In a study of fire-induced SWR, DeBano et al. [32] reported that the thickness of the hydrophobic layer increased with the decrease in clay content resulting in significantly higher water repellency in sands compared to heavy textured soils. Soil specific surface area (SA) would be a key factor controlling SWR occurrence in the pastoral systems in New Zealand. Yet, a complete assessment of the SA influence on water repellency needs data that include a wide range of SWR, soil C and SA. Hermansen et al.'s [18] survey on the South Island of New Zealand included a wide range of soil C. Still, when it comes to texture, this survey included mainly silt, silt loam and sandy loam (no clay and only one sandy textured soil) [18]. Thus, it is difficult to draw a solid conclusion on the impact of soil texture and SA on the potential SWR.

This paper aims to:

(i)　Model actual SWR persistence as a function of $\theta$ and the potential SWR in drying hydrophobic soils;
(ii)　Examine the implication of soil particle size distribution and SA in controlling the potential SWR through a combination of published datasets from New Zealand case studies.

## 2. Theory

Different models were suggested by Li et al. [21] for SWR characteristic curves for artificially hydrophobized soil (e.g., Gaussian and Lorentzian models). However, measurements of SWR persistence showed a consistent sigmoidal response to the decreasing $\theta$ in naturally hydrophobic soil. A suitable form of equations that perfectly simulate the response of SWR persistence $R_a$ to soil moisture $\theta$ (g g$^{-1}$) is the following reversed sigmoidal equation:

$$R_a(\theta) = \frac{R_p}{1 + e^{\delta(\theta - \theta_c)}}$$

(1)

where $R_p$ is the potential persistence of SWR ($R_p$ = Log WDPT(s) of dry soils) and the parameters $\theta_c$ and $\delta$ are curve characteristics that need to be determined for each soil type through fitting experimental data to the model. Factors controlling $R_p$, including soil C, texture and pH, will be discussed in Section 4.2. The first and second derivatives of Equation (1) are

$$R_a{}'(\theta) = \frac{-\delta R_p e^{\delta(\theta - \theta_c)}}{\left[1 + e^{\delta(\theta - \theta_c)}\right]^2}$$

(2)

$$R_a{}''(\theta) = \frac{\delta^2 R_p e^{\delta(\theta - \theta_c)}\left[e^{2\delta(\theta - \theta_c)} - 1\right]}{\left[1 + e^{\delta(\theta - \theta_c)}\right]^4}$$

(3)

For $\theta = \theta_c$, $R_a$ reaches its half potential, the first derivative $R_a{}'$ is in its minimum and the second derivative $R_a{}''$ is equal to 0 (inflexion point):

$$R_a(\theta_c) = \frac{R_p}{2}$$

(4)

$$R_a{}'(\theta_c) = \frac{-\delta R_p}{4}$$

(5)

$$R_a{}''(\theta_c) = 0$$

(6)

The higher the factor δ, the lower the curve slope value at the inflexion point. Thus, this parameter describes how SWR persistence changes with the drying rate. The higher δ, the faster the transition from the wettable to the water repellent state when drying soils. Parameter $\theta_c$ is moisture at the inflexion point that corresponds to half of the potential persistence. Dekker and Ritsema [7] introduced the critical water content as the value above which the soil is wettable, and below is water repellent. However, the repellency at this moisture level is easily reversible and not necessarily critical. Here, we introduce a new definition of the critical water content $\theta_c$ that corresponds to the highest increase in SWR persistence in drying soils, and it is significantly difficult to rewet hydrophobic soils past this point. Determination of $\theta_c$ is essential for the prediction of soil moisture effect on the surface runoff during rain events after dry periods.

## 3. Materials and Methods

In the present study, soil properties analysis was carried out as the following: SWR persistence and severity was measured on nine soil samples from nine pastoral sites representing four soil orders: Recent (Entisols, Inceptisols), Brown (Inceptisols), Pallic (Alfisols) and Pumice (Andisols) (New Zealand Soil Classification [33] and Soil Taxonomy equivalent [34]). Water repellency persistence vs. water content was measured on ten hydrophobic soils representing four soil orders and a wide range of textures (Table 1). The actual persistence of SWR ($R_a$) was measured using the WDPT method. The soil samples were brought to saturation, and gradually air-dried at room temperature (20 to 22 °C). Five replicates of WDPT measurements were carried out using 40 μL drops of deionized water that were placed on the soil smoothed surface, and the full drop penetration time was recorded in seconds (s) [35]. Measurement of WDPT was carried out at different levels (around 10% reduction each step) of gravimetric soil moisture (g g$^{-1}$). When moisture reached a stable minimum, air drying samples were dried at 105 °C for dry soil weight estimation. The estimation of the SWR degree was done by the MED method. Ethanol concentrations of 0, 3, 5, 8.5, 13, 24 and 36% by volume were prepared, and five droplets of 40 μL were placed on the smoothed soil sample surface. The molarity of the ethanol test was represented by the ethanol concentration of the droplet that entered the soil surface in 5 s.

To assess the effect soil particle size, SA and C content on the severity and the persistence $R_p$, we aggregated the published soil data from three recent New Zealand Studies [18,24,36]. Eight soils and their respective particle size distribution data were sampled by Whitley et al. [37] from eight dryland pasture sites, including three from the North Island and five from the South Island of New Zealand. We measured the persistence and the degree of SWR on these soils.

Observed values of the actual SWR persistence (Log WDPT) and water content were used to find the two parameters of the curve that best fit the experimental data and have the lowest RMSE:

$$\text{RMSE} = \sqrt{\frac{1}{n} \sum_{i=1}^{n} (R_{a_i} - R_{o_i})^2} \tag{7}$$

$R_{o_i}$ are the observed values of SWR persistence and $R_{a_i}$ are the fitted values using Equation (1).

The specific surface area of sand and silt fractions $a_s$, were estimated based on particles size distribution using the following equation by [38]:

$$a_s = \frac{6}{\rho_s} \sum \left( \frac{f_i}{d_i} \right) \tag{8}$$

where $f_i$ is the mass fraction of particles with a diameter $d_i$ and $\rho_s$ is the respective particle density (2.65 g cm$^{-3}$ for sand and silt and 2.67 g cm$^{-3}$ for clay were used). For clay fraction, surface area $a_c$ was estimated using Equation (9) [38]:

$$a_c = \frac{2f_c}{\rho_s l} \tag{9}$$

where $f_c$ is the mass fraction of clay particles, l is the thickness of clay platelets assumed to have an average of $4.10^{-9}$ m. Total soil specific area SA is then estimated as the sum:

$$SA = a_s + a_c \tag{10}$$

## 4. Results and Discussions

### 4.1. The Actual SWR ($R_a$) as Function Soil Moisture

The potential WDPT of air-dry soils varied between 38 and 8460 s ($R_p$ from 1.57 to 4.2) (Table 1). This means that water repellency classes varied from strong to severe according to the classification suggested by Doerr [35]. Total carbon ranged between 3.6 and 12.9 % (Table 1). Water repellency persistence (WDPT) increased significantly with decreasing soil moisture. All the studied samples representing different soil orders, textures and potential SWR represented a sigmoidal response to the decrease in soil moisture where WDPT attends a maximum for air-dry soils. This typical sigmoidal response of WDPT to moisture decrease was observed in dunes sand [39], SWR in New Zealand hydrophobic soils determined by sessile drop method [19], hydrophobic peat [40], post-fire hydrophobic soils [41] and Portuguese sandy loam and loamy sand in a forest ecosystem study [3]. However, the present result is different from data reported by Li et al. [21], who established SWR persistence curves for artificially hydrophobized soils using octadecylamine ($C_{18}H_{39}N$). Data from this study showed that the SWR persistence curve represented typical sigmoid shape for sand samples while it presented unimodal curves for loam, clay loam and silt loam samples. Results from studies using artificially hydrophobized soils cannot be generalized as a universal model for hydrophobic soil. Natural SWR is caused by a complex combination of organic materials (e.g., plants fragments, roots) [2] and hydrophobic compounds [22]. The organic materials are a principal component in high C soils. For these reasons, natural SWR is difficult to simulate by adding one hydrophobic compound to the soil. Efforts have been made to understand the reestablishment of SWR. This theory fits well with our experimental data showing a sigmoidal increase in SWR-persistence in drying soils. The present data fit well with the conceptual model of SWR development [5]. From the sigmoidal curves in the present study, three phases can be observed (Figure 1a–f). (i) The wettable phase where the hydrophobic compounds are detached from the soil minerals (suspended in water). (ii) The transition phase, which corresponds to the exposure of the mineral surfaces to the attachment of the hydrophobic compounds (from the first SWR appearance to the inflexion point). In this phase, the increase in SWR persistence is exponential. (iii) The saturation phase that extends from the inflexion point to a stable maximum. This final phase represents the coating of soil minerals that is limited by the saturation of the available surface area and the amount of the hydrophobic compounds.

After fitting the sigmoidal function (Equation (1)) to the experimental data and analysing the relationship between the three parameters of the equation, the following patterns have been observed. There was no evident relationship between the measured $R_p$ and the critical water content $\theta_c$ (Table 2). This means that the potential hydrophobicity does not control the level of moisture below; drying soils become critically water repellent (peak in the $R_a{'}(\theta)$). Nevertheless, there was a strong correlation between C and $\theta_c$ with R = 0.91 ($R^2$ = 0.82). We theorize that $\theta_c$ is logically controlled by the amount and the type of hydrophobic materials present in the C pool, as there are different compounds involved in this process [22,42]. There was a relatively low determination coefficient ($R^2$ = 0.19) for the linear regression between the $R_p$ and $\delta$ coefficient that is involved in the steepness of the slope at the inflexion point. The higher $R_p$ is, the lower the coefficient $\delta$ is and thus, the smoother the transition between the wettable and the repellent phases when drying the soils. The lower $R_p$ is, the higher $\delta$ is, and therefore, the steeper is the increase in SWR persistence when water content drops near $\theta_c$. When it comes to the relationship between $\theta_c$ and the coefficient $\delta$, there was a moderate linear relationship with $R^2$ = 0.45. The lower $\theta_c$ is, the steeper the slope is near the inflexion point (lower $\delta$) and vice versa. However, one Orthic Pumice soil diverged from this rule with a relatively low $\theta_c$ and $\delta$ (Figure 1d). This suggests that SWR persistence develop smoothly when this soil dries out, without a sudden transition toward

the potential value $R_p$. An investigation at the nanoscale is needed to understand how soil properties, such as texture and soil order, affect the hydrophobic compounds' behaviour toward soil minerals.

**Table 1.** Potential soil water repellency (SWR), C content and texture of different soil samples and the fitted model parameters with their respective RMSE. Water drop penetration time (WDPT) was measured on individual samples using five water drops at five to six decrements.

| Sample | NZ Classification | WDPT s | Texture | C % | $R_p$ Log s | $\theta_c$ g g$^{-1}$ | $\delta$ | RMSE Log s |
|--------|-------------------|--------|---------|-----|-------------|----------------------|----------|------------|
| 1 | Pallic Orthic Brown | 3900 | Clay loam | 12.89 | 3.68 | 0.35 | 18.69 | 0.18 |
| 2 | Typic Orthic Pumice | 3660 | Sand | 4.93 | 3.56 | 0.13 | 49.06 | 0.01 |
| 3 | Typic Immature Pallic | 168 | Loamy Silt | 3.6 | 2.23 | 0.13 | 57.28 | 0.00 |
| 4 | Typic Orthic Pumice | 8460 | Sand | 4.6 | 4.2 | 0.19 | 14.93 | 0.24 |
| 5 | Mottled Argillic Pallic | 782 | Silt loam | 8.65 | 2.78 | 0.26 | 33.46 | 0.11 |
| 6 | Pallic Orthic Brown | 604 | Silt loam | 6.69 | 2.78 | 0.17 | 48.84 | 0.13 |
| 7 | Mottled Argillic Pallic | 286 | Light silt loam | 7.4 | 2.42 | 0.29 | 27.75 | 0.24 |
| 8 | Pallic Orthic Brown | 604 | Silt loam | 6.69 | 2.78 | 0.21 | 37.32 | 0.22 |
| 9 | Mottled Argillic Pallic | 38 | Light silt loam | 4.92 | 1.57 | 0.16 | 33.29 | 0.14 |
| 10 | Mottled Argillic Pallic | 1800 | Light silt loam | 5.86 | 3.25 | 0.17 | 34.65 | 0.03 |

The present soils showed significantly different critical soil moisture $\theta_c$ going from 0.13 to 0.35 g g$^{-1}$ and different steepness curves at the inflexion point $\delta$, going from 14.9 to 57.3 (Table 1). Data from Dekker et al. [39] showed that the water content below which sand becomes repellent changes with soil depth. This suggests different $\theta_c$ levels over the soil profile. The decrease in $\theta_c$ would result from decreasing concentrations of hydrophobic materials in the C pool with depth if we assume that they occur at the soil surface. Indeed, hydrophobic compounds can result from plant leaves and root decomposition [2,22,43] that are probably the main source of hydrophobicity in the New Zealand pastoral systems. The potential SWR persistence is controlled by many parameters, including soil C, particle size distribution and the type and the amount of the hydrophobic materials in the C pool. In the second part of the results (Section 4.2), a detailed analysis of the implication of C and particle size on the distribution in the potential SWR is presented.

From the fitted curves of the studied soils (Table 1 and Figure 1a–f), we can differentiate three SWR persistence patterns, representing three soils categories. Soils with low $\theta_c$ (0.13 to 0.17 g g$^{-1}$) and a steep slope at $\theta_c$ ($\delta$ from 48.8 to 57.3). These soils are less susceptible to persistent SWR in the drying phase compared to soils with higher $\theta_c$. However, the $R_a$ rate increase in these soils is very high when moisture drops near $\theta_c$. This implies that a persistent water repellency can appear suddenly when water content approaches $\theta_c$ in these soils. The second category represents a low $\delta$ and high $\theta_c$. These soils express a smooth increase in SWR persistence when going from saturated soils to the critical moisture $\theta_c$. However, the high $\theta_c$ (0.26 to 0.35 g g$^{-1}$) means that these soils are more prone to SWR in the early stages during the dry periods. The third pattern was observed in the soils with the relatively higher $\theta_c$ and steepness at the inflexion point. In these soils, a persistent repellency would develop promptly in the early stages of a dry period. Remediation strategies (e.g., surfactant application) would be necessary for the second and the third categories to attenuate the agro-environmental effect of SWR [44].

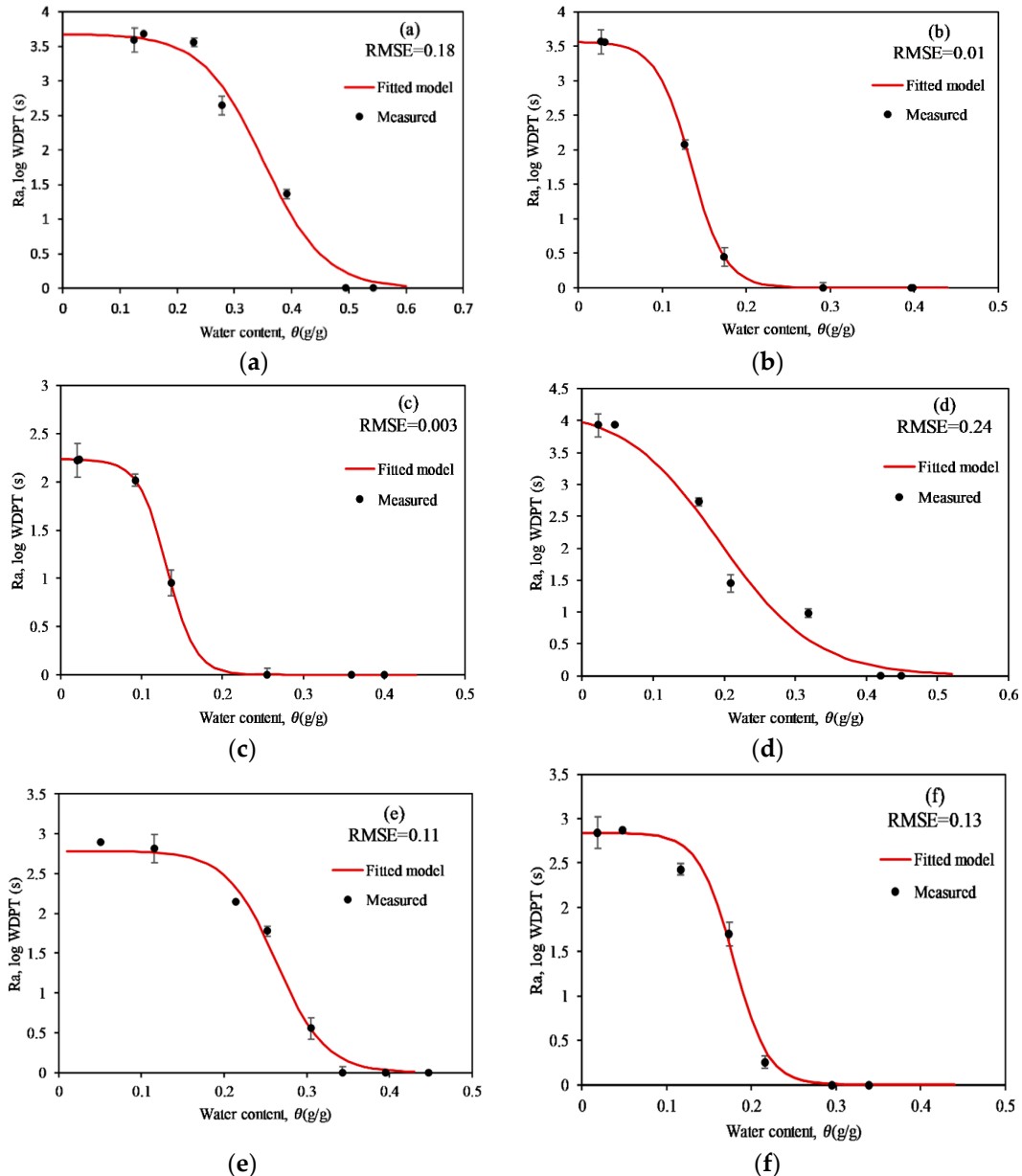

**Figure 1.** Persistence of soil water repellency (SWR) (Log water drop penetration time (WDPT) (Log s)) as a function of water content θ (g g$^{-1}$) measured during dying soil samples and corresponding fitted curve using the Equation (1) for the first six soils from Table 1: (**a**) Pallic Orthic Brown, (**b**) Typic Orthic Pumice, (**c**) Typic Immature Pallic, (**d**) Typic Orthic Pumice, (**e**) Mottled Argillic Pallic, (**f**) Pallic Orthic Brown.

**Table 2.** Coefficient of determination of linear regressions between C (%) and the model parameters $R_p$ (Log s), $\theta_c$ (g g$^{-1}$) and δ.

|  | **C** | **R$_p$** | **θ$_c$** | **δ** |
|---|---|---|---|---|
| C | 1 | 0.06 | 0.82 | 0.24 |
| **R$_p$** |  | 1 | 0.04 | 0.19 |
| **θ$_c$** |  |  | 1 | 0.45 |
| **δ** |  |  |  | 1 |

## 4.2. Soil Properties Controlling the Potential SWR ($R_p$)

Modelling the actual persistence of SWR ($R_a$) using the sigmoidal function (Equation (1)) needs a determination of the potential SWR ($R_p$). Data from a wide range of soil textures from fine (34% clay, 40% silt) to coarse (85% sand) (Table 3) showed a strong implication of soil C and soil texture in controlling $R_p$. In this combined dataset, carbon content ranged from 2.49 to 12.89% (Table 3). Soils showed different levels of SWR persistence ranging from wettable (Log WDPT = 0) to severe persistence (Log WDPT = 3.95) and a MED from 0 to 27%. Podzol and Pumice orders represented a higher severity (27 and 24%, respectively), while Ultic and Semiarid showed the lowest MED values (3 and 1.5%, respectively). Although the significant positive correlation between WDPT and MED, the reported persistence of Ultic soil from Müller et al. [24] was the highest (Log WDPT of 3.95), followed by the Pumice order (Log WDPT of 3.83). Many studies demonstrated that high persistence does not necessarily mean high severity of SWR [17].

**Table 3.** Soil characteristics from our study and other studies including pH, sand, silt, clay, C contents, surface area (SA), molarity of ethanol drop (MED) and Log WDPT.

| Soil Order (NZ) | $n$ | pH | Silt g g$^{-1}$ | Sand g g$^{-1}$ | Clay g g$^{-1}$ | SA m$^2$ g$^{-1}$ | C % | MED % | Log WDPT Log s | Study |
|---|---|---|---|---|---|---|---|---|---|---|
| Recent | 1 | 5.3 | 0.36 | 0.55 | 0.09 | 1.84 | 2.49 | 0.00 | 0.00 | |
| Brown | 1 | 5.2 | 0.39 | 0.34 | 0.28 | 5.31 | 12.89 | 13.00 | 3.73 | |
| Pumice | 1 | 5.2 | 0.14 | 0.86 | 0.01 | 0.16 | 4.93 | 24.00 | 3.77 | This study: |
| Pallic | 1 | 5.6 | 0.32 | 0.62 | 0.06 | 1.17 | 2.74 | 0.00 | 0.00 | Sampled by |
| Brown | 1 | 5 | 0.39 | 0.48 | 0.13 | 2.52 | 6.18 | 5.00 | 1.54 | Whitley et al. |
| Pallic | 1 | 5.2 | 0.47 | 0.5 | 0.03 | 0.74 | 3.6 | 13.00 | 2.84 | [37] |
| Brown | 1 | 4.7 | 0.32 | 0.51 | 0.17 | 3.27 | 4.91 | 6.75 | 1.87 | |
| Pumice | 1 | 5.1 | 0.14 | 0.85 | 0.01 | 0.18 | 6.7 | 24.00 | 3.89 | |
| Brown | 21 | 5.3 | 0.58 | 0.34 | 0.08 | 1.66 | 6.3 | 7.50 | – * | |
| Pallic | 12 | 5.5 | 0.73 | 0.21 | 0.06 | 1.30 | 3.8 | 3.00 | – | Hermansen |
| Podzol | 12 | 5.4 | 0.62 | 0.29 | 0.09 | 1.89 | 9.5 | 27.00 | – | et al. [18] |
| Recent | 18 | 5.2 | 0.67 | 0.25 | 0.08 | 1.66 | 4.2 | 9.00 | – | |
| Semiarid | 9 | 5.6 | 0.48 | 0.40 | 0.13 | 2.52 | 4.1 | 1.50 | – | |
| Pallic | 12 | 4.5 | 0.22 | 0.49 | 0.27 | 5.61 | 11.6 | 2.00 | 3.07 | Müller et al. |
| Ultic | 9 | 4.9 | 0.41 | 0.25 | 0.34 | 6.53 | 8.1 | 3.00 | 3.95 | [24] |
| Recent | 6 | 4.5 | 0.64 | 0.32 | 0.04 | 0.96 | 9.06 | 11.50 | 3.65 | Simpson et |
| Brown | 6 | 4.7 | 0.49 | 0.49 | 0.02 | 0.56 | 8.69 | 10.00 | 3.71 | al. [36] |

(*) not measured in the study.

The persistence of SWR estimated was significantly (R = 0.66; $p < 0.05$) correlated with soil C (Table 4). The simple linear regression between Log WDPT and C had $R^2$ of 0.44 and RMSE of 1.15 Log s. Using both C and SA in multiple linear regression (MLR) improved the prediction of Log WDPT ($R^2$ = 52, RMSE = 1.12 Log s) (Figure 2d).

$$\text{Log WDPT} = 0.38\,\text{C} - 0.029\,\text{SA} + 0.60 \tag{11}$$

This level of correlation is in agreement with previous results from a New Zealand North Island survey on SWR, which showed an R of 0.61 ($R^2$ = 0.37) between C and Log WDPT [26]. Hermansen et al. [18] reported a high correlation between SWR severity (measured with MED) and organic C with an R = 0.82 ($R^2$ = 0.68) in the South Island of New Zealand survey. Unlike the aggregated dataset in the present paper, soil dataset from this survey contains mainly coarse- and medium-textured soils (sand, sandy loam, silt loam). Only six soil samples out of 78 soils were silty clay and silty clay loam, and only one sand textured sample. Although MED had no significant correlation with soil properties (Table 4), C content, sand, silt, clay and SA seem to affect the MED unobtrusively. The SA was negatively correlated with MED, and C was positively correlated with MED (Table 4). As clay contributed significantly to the estimated SA (R = 1 and $p < 0.001$), there is a strong indication that there is an opposing contribution of SA and C in the expression of the severity of SWR. An increase in

the amount of the hydrophobic compounds in soil C pool enhances the coating of minerals surfaces. However, this process might be restricted by the high SA in clays.

$$\text{MED} = 1.41\,\text{C} + 161.61\,\text{sand} + 161.44\,\text{silt} + 161.04\,\text{clay} - 16148.23 \tag{12}$$

$$\text{MED} = 1.89\,\text{C} - 0.41\,\text{SA} + 4.27 \tag{13}$$

Carbon was moderately correlated with clay content and SA ($p$ = 0.028; R = 0.53) (Table 4). The estimation of SA was not 100 per cent accurate based on a coarse particle size distribution alone. In soil with high variability in particle type, using the particle size distribution can underestimate SA by one-two orders [45]. Thus, an accurate estimation of MED was possible based on C content and improved estimation of SA.

## 5. Conclusions

The suggested equation enables modelling of the actual repellency persistence ($R_a$) to the moisture decrease in drying hydrophobic soils using the potential repellency ($R_p$) and two shape parameters $\theta_c$ and $\delta$. The curve shape parameters give valuable information on how SWR persistence behaves in the drying period. In the studied hydrophobic soils, three main patterns were observed in the response curves $R_a(\theta)$. (i) The first pattern is associated with soils that have low $\theta_c$ and high $\delta$ (steep slope at $\theta_c$). These soils were less prone to express persistent SWR in the early stages of the dry period. Nevertheless, SWR persistence increased suddenly when moisture drops near $\theta_c$. (ii) The second pattern represents soils with low $\delta$ and high $\theta_c$ (0.26 to 0.35 g g$^{-1}$), which meant prompt development of SWR in the early stages of dry periods, although there was a smooth increase in SWR persistence. (iii) The third pattern showed a relatively higher $\theta_c$ and steepness at the inflexion point $\delta$. This implies that a persistent repellency would appear suddenly in the early stages of the summer period. A nanoscale investigation of soil properties controlling the SWR persistence dynamics in drying soil is needed to understand better these patterns and how they would affect the plant growth and hydrological processes.

When it comes to soil properties controlling the potential water repellency $R_p$, carbon and soil texture showed strong implications in this regard. The present data from pastoral soils showed that C has a significant influence on the potential severity of SWR and the critical soil moisture in hydrophobic soils. Both the specific surface area and soil C contribute to controlling the potential SWR degree in the studied soils. The present model and results will serve for better understanding of SWR behaviour in drying hydrophobic soils and its hydrological implications.

**Table 4.** Pearson product moment correlation matrix of pH, silt (g g$^{-1}$), sand (g g$^{-1}$), clay (g g$^{-1}$), SA (m$^2$ g$^{-1}$), C (%), MED (%), Log WDPT (Log s).

| | pH | Silt | Sand | Clay | SA | C | MED | Log WDPT |
|---|---|---|---|---|---|---|---|---|
| pH | 1 | 0.2 | −0.03 | −0.29 | −0.29 | −0.54 * | −0.01 | −0.49 |
| Silt | | 1 | −0.85 *** | −0.15 | −0.13 | −0.06 | −0.15 | 0.01 |
| Sand | | | 1 | −0.4 | −0.42 | −0.23 | 0.34 | −0.09 |
| Clay | | | | 1 | 1 *** | 0.53 * | −0.38 | 0.13 |
| SA | | | | | 1 | 0.53 * | −0.39 | 0.13 |
| C | | | | | | 1 | 0.29 | 0.66 * |
| MED | | | | | | | 1 | 0.65 * |
| Log WDPT | | | | | | | | 1 |

$p$ levels: * 0.05, ** 0.01, *** 0.001. Multiple linear regression (MLR) modelling showed that C ($p$ = 0.024), sand, silt, clay ($p$ = 0.022 each) similarly contribute at 70% of MED variation with an RMSE of 5.4% (Equation (9), Figure 2a). In contrast, an MLR model based on C and SA showed that they contributed significantly ($p$ = 0.009 and 0.005, respectively) to 48% of MED variation with an RMSE of 6.59% (Equation (13), Figure 2b,c).

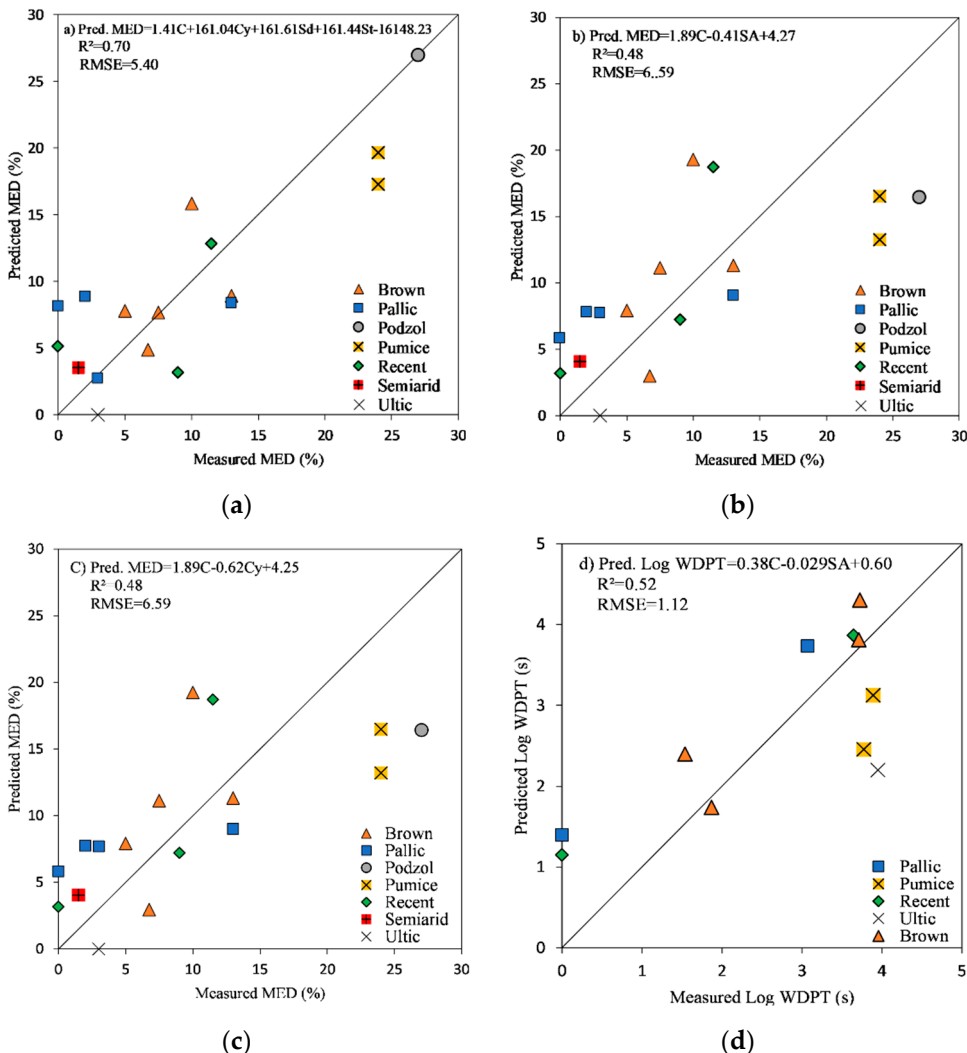

**Figure 2.** Multiple linear regressions (MLR) for the molarity of ethanol drop (MED) using (**a**) carbon and soil particles size distribution; (**b**) carbon and surface area (SA); (**c**) carbon and clay (Cy); and (**d**) the MLR for Log WDPT using C and SA.

**Author Contributions:** Writing—original draft preparation, M.B. (Mohamed Bayad); review—editing, J.M., L.C. and S.T.; supervision, H.W.C.; Resources—contribution in soil and data collection, M.B. (Moussa Bouray). All authors have read and agreed to the published version of the manuscript.

**Funding:** This work was funded by Mohammed VI Polytechnic University of Benguerir, OCP Morocco, Smart Nutrient Education Programme and Blinc Innovation, New Zealand.

**Acknowledgments:** We would like to thank Amy Whitley for providing us with nine soil samples that we used for investigating the effect of particle size distribution on SWR.

**Conflicts of Interest:** The authors declare no conflict of interest.

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
