# Peer review of "The Relationship between Soil Moisture and Soil Water Repellency Persistence in Hydrophobic Soils"

_water, doi:10.3390/w12092322_

Round 1

Reviewer 1 Report

Good work. Water replellency is a phenomenon still not very well understood, and its quantification solely depends on its measrement, and can not be predicted. As shown here, and also largely predictable, it depends on content of water repellent components such as soil organic carbon, and on surface area that depends on soil texture.

My only problem is that I can not interpret their Fig. 3. It is not clear at all what is shown here in the 3x3 panels of this Figure. Possibly, the x-axis of the left bottom graph is mislabelled, if these are correlation graphs. NOt sure though how to read the frequency charts. Is this figure really necessary, as there is no discussion of it.

Author Response

Reviewer 1:

Indeed, figure 3 represent correlation graphs for SA, C and MED. These correlations are presented in Table 3 as well. We have removed figure 3 as the Pearson correlations presented in Table 3 are enough to show the correlations between C, SA and MED.

Reviewer 2 Report

It is a very interesting work with scientific soundness and originality concerning the hydrophobic soils. The whole work is well-prepared and clearly described. My few comments are presented in the submitted pdf file.

Author Response

Reviewer 2:

89: (is),

Corrected: in

96: (Although SWR can occur in a wide range of soil textures [26,30,31], Sandy soils are more susceptible to coating by hydrophobic materials because of their low SA [32])

Corrected: Although SWR can occur in a wide range of soil textures [26,30,31], Sandy soils are more susceptible to coating by hydrophobic materials because of their low SA [32]

102: assessment

Corrected: assessment of

142: Recent, Brown, Pallic and Pumice (NZ Classification),

Corrected: Recent (Entisols, Inceptisols), Brown (Inceptisols), Pallic (Alfisols) and Pumice (Andisols) (New Zealand Soil Classification [34] and Soil Taxonomy equivalent [35])

167 and 169: Where,

Corrected: where

231: added:

is presented

All changes are tracked in the script.
